# C5aR plus MEK inhibition durably targets the tumor milieu and reveals tumor cell phagocytosis

Melissa R Perrino[1,2,*], Niousha Ahmari[1,*], Ashley Hall[1], Mark Jackson[1], Youjin Na[1], Jay Pundavela[1], Sara Szabo[3], Trent M Woodruff[4], Eva Dombi[5], Mi-Ok Kim[6], Jörg Köhl[2,7,8], Jianqiang Wu[1,2], Nancy Ratner[1,2]

Plexiform neurofibromas (PNFs) are nerve tumors caused by loss of *NF1* and dysregulation of RAS-MAPK signaling in Schwann cells. Most PNFs shrink in response to MEK inhibition, but targets with increased and durable effects are needed. We identified the anaphylatoxin C5a as increased in PNFs and expressed largely by PNF macrophages. We defined pharmacokinetic and immuno-modulatory properties of a C5aR1/2 antagonist and tested if peptide antagonists augment the effects of MEK inhibition. MEK inhibition recruited C5AR1 to the macrophage surface; short-term inhibition of C5aR elevated macrophage apoptosis and Schwann cell death, without affecting MEK-induced tumor shrinkage. PNF macrophages lacking C5aR1 increased the engulfment of dying Schwann cells, allowing their visualization. Halting combination therapy resulted in altered T-cell distribution, elevated Iba1+ and CD169+ immunoreactivity, and profoundly altered cytokine expression, but not sustained trumor shrinkage. Thus, C5aRA inhibition independently induces macrophage cell death and causes sustained and durable effects on the PNF microenvironment.

## Introduction

The tumor microenvironment is a key promoter of tumorigenesis (Hannahan & Weinberg, 2011). Peripheral nerve tumors caused by loss of the *NF1* tumor suppressor provide a unique system in which to study the roles of the tumor microenvironment in the formation of benign tumor growth. This is because, whereas neurofibroma Schwann cells (SC and/or their precursors) are the crucial pathogenic cells in neurofibroma and the only cells containing biallelic *NF1* mutations (Serra et al, 2001; Pemov et al, 2017), plexiform neurofibromas (PNF) also contain mast cells, dendritic cells, endothelial cells, T cells, fibroblasts, perineural-like cells, and macrophages. PNF macrophages are the dominant immune cell in PNF, accounting for at least half of the tumor immune mulieu. Interplay among tumor and immune cells is believed to play important roles in neurofibroma formation and growth (Fletcher et al, 2020; Jiang et al, 2021). For example, loss of *Nf1* enhances inflammatory cytokine expression in cultured SC (Yang et al, 2003), and injury-associated inflammation facilitates neurofibroma development in mouse models (Rizvi et al, 2002; Staser et al, 2012; Ribeiro et al, 2013). In human and murine PNF, macrophages account for 30% of tumor cells, accompanied by dendritic cells and T cells with variable expression of T-cell activation and exhaustion markers (Farschtschi et al, 2016; Torres et al, 2016; Haworth et al, 2017).

Of individuals with NF1, up to 25–30% have visible or symptomatic PNF and >50% have one or more MRI-detectable PNF that develops in early life and grows most rapidly in the first decade of life (Dombi et al, 2007; Mautner et al, 2008; Prada et al, 2013). PNF are associated with nerve trunks and can cause substantial morbidity including pain, neurologic deficits, and motor dysfunction. No cure exists for PNF apart from surgical removal (Canavese & Krajbich, 2011). Consequently, substantial effort is directed toward identification of therapy for these lesions. These tumors are irregular in shape, and tumor volume changes are variable, so PNF volume change measured over time in three dimensions (3D) is currently recommended for use in NF1 PNF clinical trials (Dombi et al, 2013).

*NF1* is a tumor suppressor gene that encodes neurofibromin, a Ras-GTPase activating protein (Ras-GAP) that augments the intrinsic GAP activity of RAS proteins (McCormick, 1995; Le & Parada, 2007). Therefore, mutation/loss of *NF1* increases signaling through the RAS pathway including the RAF/MEK/ERK signaling cascade. MEK inhibition (MEKi) shrinks 70% of tumors in *DhhCre;Nf1[fl/fl]* neurofibroma-bearing mice (Wu et al, 2008; Jessen et al, 2013; Jousma et al, 2015). Phase I/II clinical trials for children with NF1 and large inoperable plexiform neurofibroma using the MEKi Selumetinib showed that 70% of patients similarly experienced sustained

[1]Division of Experimental Hematology and Cancer Biology, Cincinnati Children's Hospital Medical Center, Cincinnati, OH, USA  [2]Department of Pediatrics, University of Cincinnati, Cincinnati, OH, USA  [3]Departmentd of Pediatrics and Pediatric Pathology, Cincinnati Children's Hospital Medical Center, University of Cincinnati, Cincinnati, OH, USA  [4]School of Biomedical Sciences, The University of Queensland, St Lucia, Australia  [5]Pediatric Oncology Branch, National Cancer Institute, Bethesda, MD, USA  [6]Department Biostatistics, University of California, San Francisco, CA, USA  [7]Institute for Systemic Inflammation Research, Lübeck, Germany  [8]Division of Immunobiology, Cincinnati Children's Hospital Medical Center, Cincinnati, OH, USA

Correspondence: Nancy.ratner@cchmc.org; Jianqiang.Wu@cchmc.org
*Melissa R Perrino and Niousha Ahmari contributed equally to this work

partial responses (Dombi et al, 2016; Gross et al, 2020). Yet, an important limitation of MEKi treatment in NF1 patients is the regrowth of plexiform neurofibroma after decreasing the treatment dose or stopping treatment. Moreover, maximal tumor volume decrease was 50% even in the best responders (Dombi et al, 2016; Gross et al, 2020). Therefore, there is an urgent need to identify targetable pathways that result in more profound and/or durable tumor shrinkage.

MEK inhibition is known to directly alter tumor immune composition and function in other tumor models (Baumann et al, 2020; Verma et al, 2021). We postulated that targeting the microenvironment together with MEKi might be a promising therapeutic direction and searched for inflammatory targets in PNF which are unaffected by MEKi treatment of PNF. Here we focus on C5a, the smaller cleavage fragment of the complement factor C5, which is a pro-inflammatory mediator. C5a is a chemoattractant for multiple inflammatory cell types, stimulates cytokine and chemokine release, and increases vascular permeability (Kolev et al, 2014; Laumonnier et al, 2017). C5a deposition was a feature of early benign tumor formation in a skin tumor model (Medler et al, 2018). At low levels, C5a can inhibit tumor growth through stimulation of innate immune cells and halt tumor cell cycle progression. At higher concentrations, C5a accelerates tumor progression by promoting an immunosuppressive microenvironment, inducing angiogenesis, and/or increasing the motility of cancer cells (Corrales et al, 2012; Kolev & Markiewski, 2018; Magrini et al, 2022).

C5a binds two G-protein-coupled receptors, C5a receptor 1 (C5aR1/CD88) (Gerard & Gerard, 1991) and C5a receptor 2 (C5aR2/C5L2) (Cain & Monk, 2002; Okinaga et al, 2003). The C5a pseudo-irreversible C-terminal cyclic peptide PMX205 Cyclo (N2-[Oxo-3phenylpropy]-Orn-Pro-D-Cha-Trp-Arg) blocks C5aR1. PMX205 was successfully used as a single agent in animal models of inflammation and recently in a carcinoma model (Paczkowski et al, 1999; Seow et al, 2016; Kumar et al, 2018; Medler et al, 2018; Ding et al, 2020). In a model of sepsis, however, only targeting both C5aR1 and C5aR2 with the C5a mutant peptide A8$^{\Delta71-73}$ improved mouse survival (Otto et al, 2004; Rittirsch et al, 2008). Targeting C5aR1 and C5aR2 with A8$^{\Delta71-73}$ also increased survival and reversed disease signs in an experimental Gaucher disease model (Pandey et al, 2017).

Here we identify C5a as elevated in neurofibroma and find that cell surface localization of C5aR1 is elevated by MEKi. We compared single-agent and combination effects of the two peptide C5aR antagonists in a murine model of PNF. We find that both genetic knockdown of C5aR1 and C5aR antagonists promote macrophage apoptosis. In addition, we find that combination effects of C5aR antagonists in combination with MEK inhibition on the tumor microenvironment and on PNF cell proliferation persist after drugs are withdrawn.

# Results

## C5a is present in PNF and C5aR-expressing macrophages are increased in PNF

Neurofibroma lysates show elevated levels of inflammatory cytokines/growth factors as compared to normal nerve/dorsal root ganglion control lysates (Choi et al, 2017). We used cytokine arrays to compare levels of immune regulatory factors in murine PNF from mice treated with vehicle or MEK inhibitor MEKi, PD0325901 (1.5 mg/kg/d; o.g.; 5 d on, 2 d off treatment regimen) for 60 d. Notably, MEKi therapy decreased expression of the most analyzed inflammatory cytokines/growth factors (Figs 1A and S1). However, C5/C5a and IL-16 remained unchanged by MEK inhibition in tumor lysates (Fig 1A). We reasoned that these factors might be druggable targets, independent of the RAS/MEK/ERK pathway. Here, we focused on C5a and its receptors.

In cultured primary human SCs from normal nerve or neurofibroma, expression of mRNA encoding C5 and C5aR1/R2 was low; C5 and C5aR1/R2 mRNAs were present at higher levels in human nerve and human PNF tissue samples, indicating that complement and complement receptor expression is largely in cells of the tumor microenvironment (Fig 1B). Immunohistochemical staining of C5aR1/CD88 in human samples increased expression in neurofibroma versus nerve (Fig 1C). To define neurofibroma cells expressing C5aR1/CD88, we turned to a mouse model. Multi-parametric flow cytometry of cells dissociated from paraspinal $Nf1^{fl/fl};DhhCre$ PNF (Wu et al, 2008) or normal nerve/Dorsal root ganglia (DRG) (WT) showed that C5aR1 is expressed on nearly 40% of macrophages and 5–10% of dendritic cells in WT and PNF (of live CD45$^+$ cells), and that the number of C5aR1+ cells increases in tumor largely because of the known increased numbers of immune cells in tumors and not an increase in C5aR1 protein on macrophages (Fig 1D–F). Consistent with previous studies, macrophages, the primary cell type expressing C5aR1, are significantly increased in PNF (Fig 1G). C5aR1 levels were similar in macrophages expressing immune-suppressive markers (CX3CR1+ or CD163+MHCII–) and in dendritic cells (CD11c+; CD11b+ or CD11b–) using the panel of antibodies (Figs 1G and S2A) and gating strategy (Fig S2B). Indeed, no significant increase in C5aR1 expression was observed in any immune syptypes in PNF versus WT nerve cells (Fig S1A and B). Thus, the increase in C5aR1 in murine tumors is because of immune-cell expansion in PNF. We generated a sunburst plot to visualize proportions of PNF cells expressing C5aR1; only immune cells expressed C5aR1 in PNF (Fig 1H). Some undefined C5aR1+ myeloid cells were CD11b+ SSC$^{hi}$ and are not shown in the plot. At least 95% of C5aR1+ myeloid cells were CD11b+; SSC$^{lo}$;F480+; CD14$^{hi}$; SERPa$^{hi}$ macrophages.

## Genetic depletion of C5aR1 does not affect neurofibroma number or size

To test if whole-body removal of the dominant C5aR, C5aR1, would affect peripheral nerve or PNF formation, we mated $DhhCre;Nf1^{fl/fl}$ mice to $C5aR1^{-/-}$ mice and compared $Nf1^{fl/fl};DhhCre$ mice to $Nf1^{fl/fl};DhhCre;C5aR1^{+/-}$ and $Nf1^{fl/fl};DhhCre\ C5aR1^{-/-}$ littermates. Neither genetic reduction nor ablation of C5aR1 altered mouse survival, tumor volume, or tumor number (Fig 2A–C). Thus, toxicity is unlikely to develop from long-term use of C5aR1 inhibitors in this setting. We then used flow cytometry to analyze effects of C5aR1 loss in the immune cells of PNF. Genetic deletion of C5aR1 did not alter total immune cells, macrophages, or dendritic cells (Fig 2D and E).

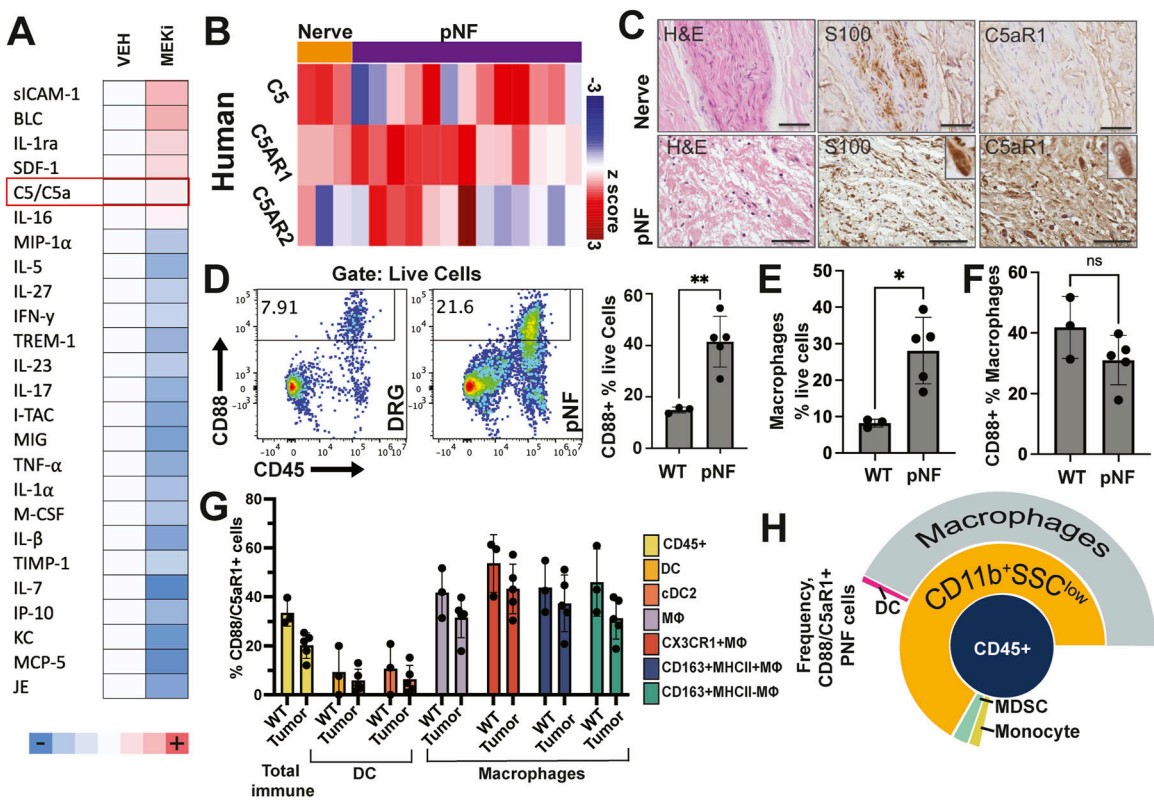

**Figure 1. The inflammatory protein C5a is expressed in mouse and human neurofibromas; C5a receptors are expressed mainly on nerve macrophages.**
**(A)** Cytokine expression in mouse PNF lysates collected from PNF-bearing mice treated for 60 d with either vehicle (control) or MEKi (PD0325901). Pixel intensity for each factor was normalized to values vehicle treated controls. C5a expression (red) was unchanged, whereas most other factors were reduced or increased by MEKi treatment. **(B)** Heatmap of C5, C5AR1, and C5AR2 mRNAs, showing variable expression in normal nerve (N) and neurofibroma (PNF). Z-Score scale shown. **(C)** Paraffin sections of human normal nerve (top) and human PNF (bottom) stained with hematoxylin and eosin (H&E), anti-S100b (brown), or anti-C5aR1 (brown). Immunohistochemistry shows increased C5aR1 protein expression in PNF. Scale bars = 50 $\mu$m. **(D)** Flow cytometry shows increased numbers of live CD88/C5aR1+ cells in mouse PNF versus DRG/nerve. **(E, F, G)** Quantification of flow cytometry data. CD88/C5aR1+ macrophages (gated as Live, CD45+; TCRb−;CD11b+; SSC$^{low}$; CD14++; CD172a++; Ly6C−;Ly6G−;F480+) are increased in pNF (* = $P$ < 0.05 by $t$ test; each data point shows data from a single mouse (n = 3–6/condition). **(E)** Total macrophages are increased as a percent of live cells in PNF. **(F)** Similar percentages of macrophages express C5aR1 in DRG/nerve and PNF. **(G, H)** Percentages of subtypes of immune cells that express CD88/C5aR1+ in DRG/nerve and PNF shown as a bar chart (G) and cell their frequency in PNF (H).

## MEK inhibition increases cell surface C5AR1 on macrophages

We treated tumor-bearing DhhCre; $Nf1^{fl/fl}$ mice with MEKi for 5 d. This pharmacologic blockade did not significantly alter the percentages of C5aR1+ tumor macrophages (Fig 2F). We next tested whether MEK inhibition alters cell surface expression of C5aR1 in tumor macrophages. We treated BMD-macrophages from tumor-bearing mice for 1 h ex vivo with MEK inhibitor or vehicle; MEKi significantly increased the intensity of anti-C5aR1/CD88 on tumor macrophage cell surfaces (Fig 2G). Imaging flow cytometry verified an increase in the fluorescence indicating C5aR1 on the tumor macrophage cell surface rather than the numbers of cells expressing C5aR1 (Figs 2H and S3A).

## Short-term combined C5aRA inhibition + MEK inhibition induces macrophage apoptosis

To test the effects of inhibition of the C5a-driven complement cascade in PNF, we used two C5aR peptide antagonists. The C5a C-terminal cyclic peptide PMX205 targets C5aR1 (Paczkowski et al,

1999; Seow et al, 2016; Kumar et al, 2018), and the A8$^{\Delta71–73}$ peptide targets C5aR1 and C5aR2 (Otto et al, 2004). To determine if co-targeting C5aR and MEK shows effects different from MEK inhibition alone, we treated $Nf1^{fl/fl}$;DhhCre neurofibroma-bearing mice 5 d with vehicle (methylcellulose), MEKi (PD0325901), C5aRA (PMX205 or A8$^{\Delta71–73}$), or C5aRA + MEKi. Cytokine protein arrays on tumor lysates showed that combination therapy reduced inflammatory proteins similar to or more than other treatment groups (Fig 3A). In contrast, the combination treatment increased apoptotic cell death, shown as cleaved PARP in tumor lysates (Fig 3B).

To verify that the combination affects the tumor microenvironment, we returned to genetic depletion of C5aR1. We compared $Nf1^{fl/fl}$;DhhCre mice to $Nf1^{fl/fl}$;DhhCre;C5aR1$^{+/−}$ and $Nf1^{fl/fl}$;DhhCre; C5aR1$^{−/−}$ littermates treated with MEK inhibitor for 5 d. PNF from these mice were dissected, cells dissociated, and cells subjected to flow cytometry. At this early time point, MEK inhibition alone had little effect on death of cells isolated from murine PNF. In contrast, genetic deletion of C5aR1 significantly increased CC3+ cells (Fig 3C, left) using appropriate gating (Fig S3B). The combination of C5aR1 deletion and MEK inhibition did not alter the amount of cell death.

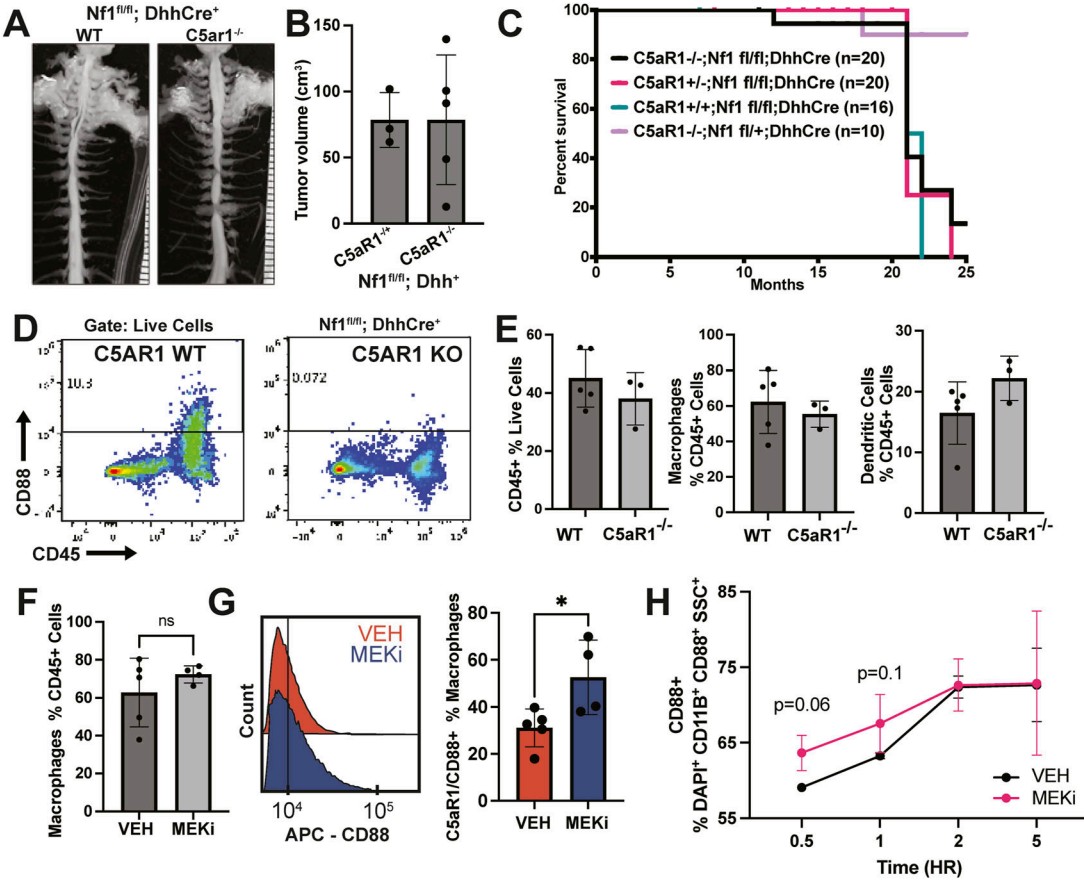

**Figure 2. Genetic depletion of C5aR1 does not alter neurofibroma formation, but MEK inhibition (MEKi) increases cell surface CD88/C5aR1 expression on PNF macrophages.**
**(A)** Gross dissections of representative *Nf1*<sup>fl/fl</sup>;*DhhCre* mouse spinal cords with attached DRG/nerves from *C5aR1*<sup>+/−</sup> (WT) and *C5aR1*<sup>−/−</sup> littermates, which form neurofibromas. **(B)** Tumor volume (cm³) measured by volumetric MRI in 8-mo-old *Nf1*<sup>fl/fl</sup>;*DhhCre*;*C5aR1*<sup>+/−</sup> (WT) and *Nf1*<sup>fl/fl</sup>;*DhhCre*;*C5aR1*<sup>−/−</sup> neurofibroma-bearing mice. Each point shows data from a single mouse. **(C)** Kaplan-Meier curve demonstrates that neither genetic reduction nor ablation of C5aR1 alters survival of *Nf1*<sup>fl/fl</sup>;*DhhCre* mice. **(D)** Representative flow cytometry showing the absence of C5aR1 in immune cells of C5aR1 null mice versus WT mice. **(E)** Flow cytometry shows that genetic deletion of C5aR1 does not alter total immune (live; CD45⁺) cell, macrophages (live; CD45⁺; TCRb−; CD11b+ SSClow; Ly6c/ly6G−; CD14+++; CD172a++), or dendritic cells (live; CD45⁺; TCRb−; CD11c+; MHCII+). **(F)** *Nf1*<sup>fl/fl</sup>;*DhhCre* mice treated with vehicle (VEH) or MEKi for 5 d do not alter percentages of tumor macrophages expressing C5aR1/CD88. Each point shows data from one tumor. (* = *P* < 0.05 by *t* test). **(G, F)** Intensity of C5aR1 on tumor macrophages increases after 5 d of MEKi treatment; histogram at right shows concatenated data from samples in (F), quantifying C5aR1 intensity. **(H)** Imaging cytometry of BMD-macrophages from tumor-bearing mice treated for 1 h ex vivo with vehicle or MEK inhibitor, showing an increase in cell surface CD88 after MEK inhibition.

Cell proliferation (Ki67+ cells), reduced by MEK inhibition, was also low in combination-treated PNF (Fig 3C, right). Gating on PNF macrophages showed that the increased cell death could be ascribed largely to dying macrophages (Fig 3D).

### Schwann cell apoptosis in PNF

Some S100b+ Schwann cells in PNF from DhhCre;*Nf1*<sup>fl/fl</sup>;*C5aR1*<sup>−/−</sup> mice were also apoptotic (CC3+) (Fig 3E). Strikingly, however, Schwan cell death was no longer detectable when MEK inhibitor was administered to DhhCre;*Nf1*<sup>fl/fl</sup>;*C5aR1*<sup>−/−</sup> mice for 5 d. Thus, PNF in DhhCre;*C5aR1*<sup>−/−</sup> show significantly increased macrophage apoptosis, with or without MEK inhibition. Schwann cell death is detectable only on loss of C5aR1 and obscured or largely prevented by MEK inhibition. To confirm these in vivo effects of C5aR1 and MEK inhibitors on PNF macrophages and Schwann cells, we stained tissue sections to detect TUNEL+ cells (Fig 3F). We verified increased

death in tumors from mice treated with A8<sup>Δ71−73</sup> peptide and with MEKi combinations with each of two C5aR1 antagonist peptides (Fig 3F and G). Therefore, inhibition of C5aR1 function promotes macrophage and Schwann cell apoptosis in PNF, and Schwann cell apoptosis is reduced by concurrent MEK inhibition.

### C5aRA + MEKi reduces tumor size, similar to MEKi alone

As the PK properties of the A8<sup>Δ71−73,</sup> peptide, a C5a mutant lacking amino acids 71–73 and harboring D69R and C27A replacements, had not been reported, we developed tested how long it persists in plasma. We found that, like C5a itself, the peptide has a very short half-life in circulation. After a single i.p. injection into WT mice, maximum concentrations (4.5 μg/ml [0.64 μM]) were detected in plasma at 30 min, which exceeds the in vitro effective dose of 0.1 μM (Fig 4A). In plasma from PNF-bearing mice treated for 5 d with

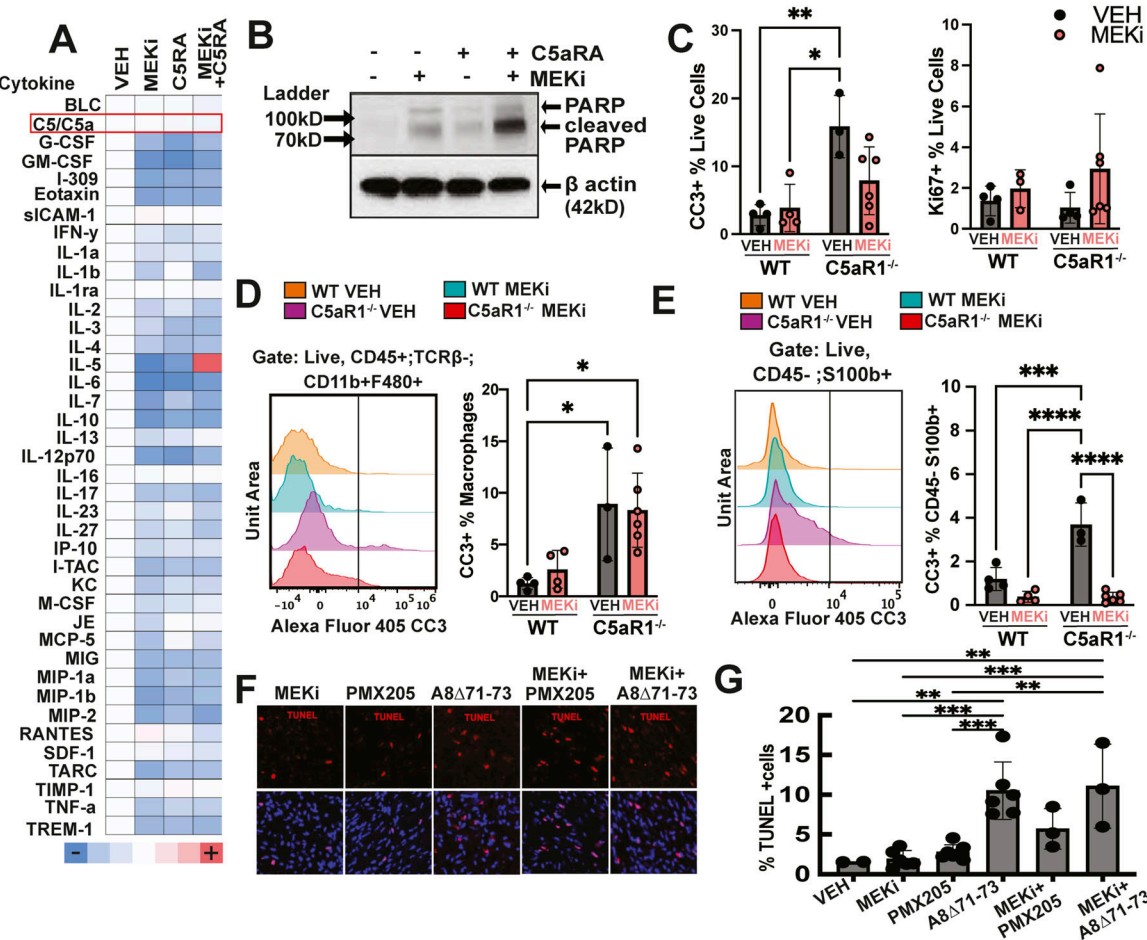

**Figure 3. Reducing C5aR1 in combination with short-term MEK inhibition (MEKi) reduces PNF cell proliferation and induces PNF cell apoptosis.**
**(A)** Heatmap shows similar changes in immune proteins by treatment for 5 d with PD0325901 (MEKi), A8$^{\Delta71-73}$ (C5AR1/2 antagonist), or the A8$^{\Delta71-73}$ + PD0325901 combination. C5/C5a is indicated in red and is unchanged. **(B)** Western blot shows increased apoptotic cell death (cleavage of PARP) in PNF lysates from mice treated with vehicle, MEKi, C5aRA, or the combination collected 2 h a final dose (pooled lysates, n = 3 mice per condition). **(C, D, E)** Flow cytometry of tumor from C5aR1-null or WT neurofibroma-bearing mice. In (C), increase percentages of cleaved caspase 3+ cells (CC3, apoptosis); percentages of proliferative cells (Ki67+ cells) were unchanged. **(D)** Histogram of concatenated (pooled) samples (left) and quantification (right) shows an increase in cleaved caspase 3 (CC3) in tumor macrophages lacking C5aR1, with no additional effect after short-term (5 d) MEK inhibition. **(E)** Histogram of concatenated (pooled) samples (left) and quantification (right) showing an increase in CC3+ in Schwann cells (CD45⁻; S100b+), which was prevented in C5aR1 null mice by 5 d of MEKi treatment. Each point shows data from one mouse (* = P < 0.05, two-way ANOVA). **(F, G)** Pharmacologic inhibition of C5aR results in cell death. **(F)** TUNEL staining of PNF sections demonstrates increased cell death in C5aRA and combination groups. **(G)** Quantification of percentage of TUNEL+/DAPI+ cells; each dot shows data from a single mouse, 3–5 hpf/mouse. **P = <0.01, ***P = <0.001; Tukey's multiple comparisons ANOVA.

A8$^{\Delta71-73}$, we detected 0.6 ng/ml A8$^{\Delta71-73}$, suggesting no significant difference from WT mice (not shown).

We then treated PNF-bearing mice with MEKi, C5aRA (PMX205 or A8$^{\Delta71-73}$ separately) alone, or the combination for 60 d, to test if combination effects persist over time and/or if we identified changes in tumor volume. The combination therapy showed no significant toxicity; mice survived and did not show morbidity or significant weight loss, defined as change in weight >10% (Fig 4B). MRI was performed for volumetric assessment of tumor growth. To obtain pre-treatment growth trajectories, MRIs were obtained at 5 mo of age and 7 mo. Mice were treated from 7 to 9 mo, when a post-treatment MRI was obtained. Waterfall plots of tumor volume and representative MRI images are shown in Fig 4C. Representative MRI images are shown in Fig 4D. Similar to results from previous studies in this preclinical model PD0325901 treatment shrank PNF in

72.7% of mice (Jousma 2015; Dombi et al, 2016). PNF in mice treated with C5aRA (PMX205 or A8$^{\Delta71-73}$); the difference between PMX205 and PD0325901, in combination and PD0325901, however, was not significant (P = 0.6979). Mice treated with A8$^{\Delta71-73}$ and PD0325901 in combination had the best apparent response, with 84% of mice having tumors smaller at 9 mo than at 7 mo, and 24% of mice with tumors smaller than the initial MRI at 5 mo. However, even by this measure, the A8$^{\Delta71-73}$ and PD0325901 combination was not significantly different from treatment versus PD0325901 alone (P = 0.6499). Mixed model analysis, which accounts for the differences in tumor growth rate in each mouse and differences in growth of tumors over time, confirmed effects of PD0325901 and each combination treatment group. A8$^{\Delta71-73}$ alone reduced tumor growth rate compared to vehicle (−0.1319, SE 0.04156, P = 0.0020); the reduction was less than the reduction caused by MEKi alone. PMX205 alone

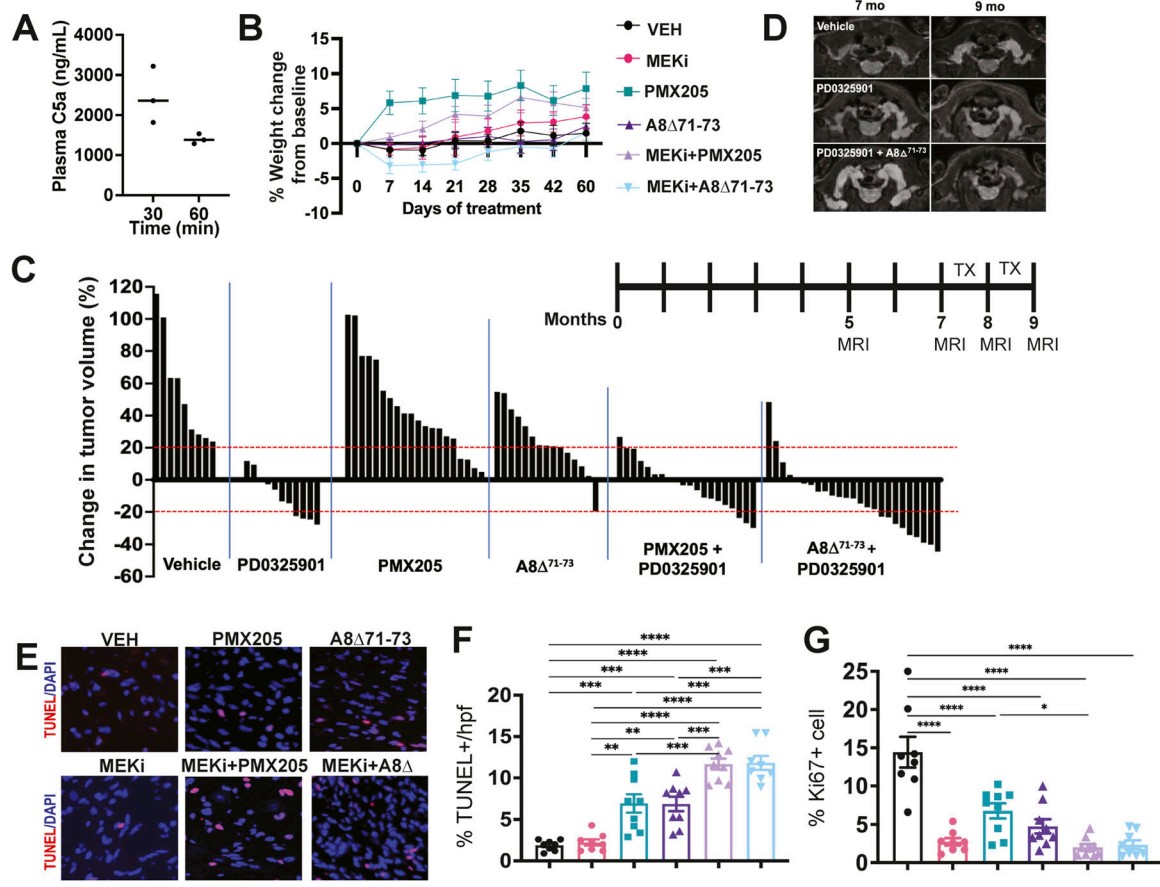

**Figure 4. 60 d of combination treatment reduces PNF volume and PNF cell proliferation and increases PNF apoptotic cell death.**
**(A)** Pharmacokinetic profile of hC5a A8$^{\Delta71-73}$ in plasma following a single i.p. administration of human C5 A8$^{\Delta71-73}$ (5 mg/kg in WT mice). Time after dosing is shown on the x axis. Each point represents data from one mouse. **(B)** Change in weight (%) by treatment group Inset shows no significant change in body weight after 60 d of treatment. **(C)** Waterfall plot showing change in tumor volume among treatment groups between 7–9 mo. of age (treatment period). Each bar shows data from a single mouse. **(D)** Representative MRI images of tumor volume change after treatment in select animals. **(C, E, F)** TUNEL staining by immunofluorescence (red) showing apoptotic cell death in treatment groups after 60 d of treatment in sections from animals from (C). **(F)** Quantification of dying cells (TUNEL staining). **$P$ = <0.01; ***$P$ = <0.001; ****$P$ =<0.0001 by Tukey's multiple comparison test. **(G)** Quantification of cell proliferation (percent Ki67+ cells) by immunofluorescence shows decreased proliferation in all treatment groups. ****$P$ = <0.0001 by Tukey's multiple comparisons test. *$P$ = <0.01 by Tukey's multiple comparison test.

did not significantly reduce tumor growth rate compared to vehicle-treated mice. We conclude that MEK inhibition alone or in combination with either C5aRA is effective in decreasing overall tumor size and that there is no significant difference among the groups.

Next, we whether chronic (60 d) treatment of *Nf1$^{fl/fl}$;DhhCre* PNF with single agent or combination therapy mimics the apoptosis observed at 5 d of treatment. Tissue sections from vehicle and PD0325901 PNF demonstrated minimal apoptosis (0.83% and 1.78%, respectively), as previously demonstrated in this model (Jessen et al, 2013). Apoptosis increased in all C5aRA treatment groups; combination therapy with PD0325901 and C5aRA showed a significant increase in TUNEL staining compared to vehicle or PD0325901 ([$P$ = <0.0001 each] Fig 4E and F). Apoptosis was not significantly different between the combination groups, showing no advantage of PMX205 versus A8$^{\Delta71-73}$ in this measure (PMX205 + PD0325901 11.67%, A8$^{\Delta71-73}$ + PD0325901 11.81%, $P$ = >0.99). Proliferation remained low in all treatment groups ($P$ = <0.0001 compared to vehicle [Fig 4G]), indicating that combination treatment does not

alter the MEK-inhibitor-driven reduction in cell proliferation in this model.

We wondered if C5a levels might change on treatment, which would be a useful biomarker to follow patient response to therapy. C5a levels were measured in the tumor after excision with a novel sandwich ELISA, but did not show a statistical differences among treatment groups (Fig S4A). Thus, C5a is not a direct readout of efficacy and on-target effect by C5AR1 antagonism.

## Both C5aR1 inhibition controls macrophage clearance of dying schwann cells

Macrophages play crucial roles in clearing dead cells, and it is believed that their efficient clearance capacity contributes to the infrequent observation of apoptotic cells in tissues with high cell turnover. Recent studies, including Baumann et al (2020), have demonstrated that MEK inhibitors can enhance the phagocytic activity of macrophages. To investigate whether MEK inhibitor enhances phagocytosis in macrophages derived from mice with

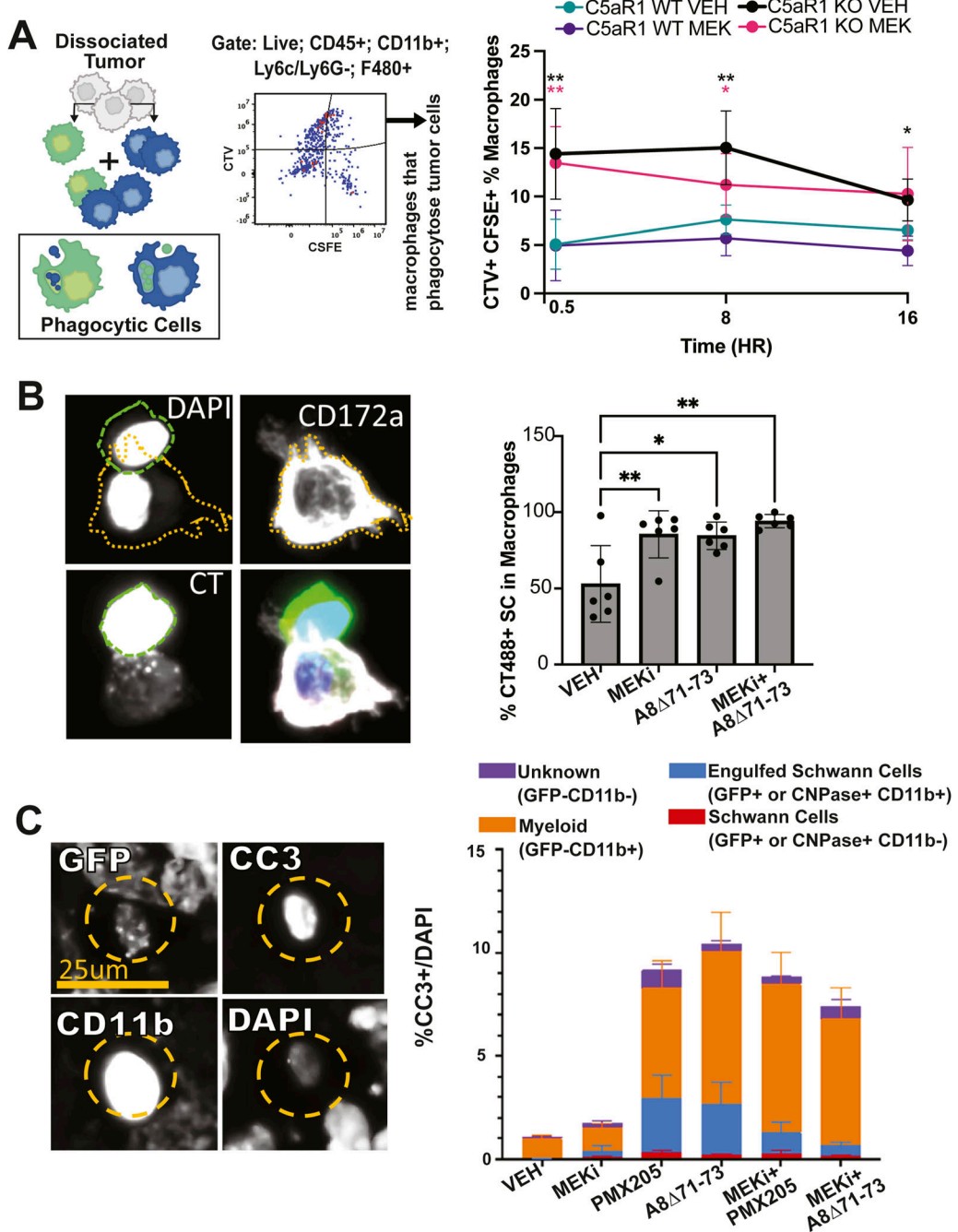

**Figure 5. Phagocytosis of Schwann cell is increased by pharmacologic C5aR blockade or MEKi.**
**(A)** Schematic at left shows mixing of labeled cells from PNF to enable detection of cell phagocytosis. Middle, representative flow cytometric result indicating quadrant containing violet and green single live cells (e.g., phagocytosis). Right, Quantification at designated times showing that MEKi reduces detection of phagocytosed cells. **(B)** Photomicrograph shows a representative image of BMDM In vitro, identified using CD172a/SIRPa immunoreactivity (white), engulfing a CFSE-labeled embryonic $Nf1^{-/-}$ Schwann cell (green) (left). At right, quantification (as defined by CSFE signal in macrophages) showing increased Schwann cell phagocytosis by both inhibitors and combination (* = P < 0.05, three technical and two experimental replicates one-way ANOVA). **(C)** Immunofluorescent co-labeling of cells in tissue sections from mice in Fig 4C, stained with anti-CNPase (Schwann cells) or DhhCre; $Nf1^{fl/fl}$-lineage traced GFP+ cells (Schwann cells), anti-CD11b (macrophages and some dendritic cells), and cleaved caspase 3 (CC3, apoptosis). Insets (left) highlight an apoptotic Schwann cell being phagocytized by a macrophage after 60 d of treatment. Right, quantification of cell death in a stacked bar graph subclassified by types of cells dying. Each data point was generated from an average 10 hpfs/PNF, in mice n = 7 mice/condition.

PNF, we isolated cells from PNFs of WT and C5aR1−/− mice and labeled them with Cell Trace Violet or CFSE (green). Violet and green cells were then mixed and treated with vehicle or MEK inhibitor for 8 h in vitro. To measure phagocytosis, post-incubation, we sorted macrophages and quantified the number of cells labeled with both dyes (Fig 5A). The absence of C5aR1 led to an increase in phagocytic

activity, but MEK inhibition did not. To test if there are Schwann cell-specific effects on phagocytosis, we removed BMDM from tumor-bearing mice incubated them with CFSE-labeled embryonic $Nf1^{-/-}$ Schwann cells. CD172a+ macrophages contained CFSE+ cells, as depicted in Fig 5B, verifying phagocytic activity. After 8 h, the inhibition of either MEK or C5a, or the combination, doubled numbers of macrophages containing CFSE+ Schwann cells. Thus, inhibiting either MEK or C5aR, individually or in combination, enhances the phagocytosis of mutant Schwann cells by myeloid cells, whereas C5aR selectively increases phagocytosis by tumor macrophages.

To determine if cell phagocytosis is altered by inhibitor treatment in vivo, we treated $Nf1^{fl/fl};DhhCre$, β-actin-LSL-GFP tumor-bearing mice, in which GFP serves as a Schwann cell lineage marker with MEKi (PD0325901) alone, C5aRA (PMX205 or A8$^{Δ71-73}$) alone, or the combination of MEKi and C5aRA for 60 d. We co-labeled cells with cleaved caspase 3 (CC3) and the myeloid marker anti-CD11b, or anti-GFP; in GFP-animals, we used CNPase as a Schwann cell marker with similar results. Indicating Schwann cells undergoing apoptosis and being engulfed by macrophages, we identified CC3+ puncta and CNPase+ (or GFP+) within CD11b+ cells (Fig 5C, left). We detected low levels of Schwann cell death, with CC3+GFP+ cells detected in C5aRA-treated mice and combination treatment arms, but negligible amounts of Schwann cell death in mice treated with vehicle or MEKi alone (Fig 5C, right). In all treatment groups, most of the CC3+ cells were CD11b+ myeloid cells. The amount of cell death marked CC3+ cells was similar to that shown by TUNEL (Fig 4), consistent with tumor cells dying by apoptosis. Thus, in vivo blockade of C5aR(s) enables detection of ongoing Schwann cell apoptosis in PNF. These results confirm those shown in Fig 3E, in which a significant increase in dying Schwann cells was detected in PNF-bearing C5aR1 null mice but reduced or lost in mice treated with MEKi.

### C5aRA + MEKi alters the microenvironment, increasing an anti-tumor phenotype

Continued administration of MEKi is necessary to prevent tumor regrowth, i.e., the effect is not durable (Jousma et al, 2015; Dombi et al, 2016). To test if combination therapy provides more durable effects than MEKi alone, we treated mice for 1 mo with A8$^{Δ71-73}$ and PD0325901, and then maintained mice off therapy for 1 mo (durability cohort) (schema, Fig 6A). After 1 mo off drug, the tumors showed increased size based on MRI scans (Fig 6B). The imaging characteristics of these neurofibromas were not notably altered from the on-treatment cohorts. In spite of increased tumor size, the numbers of Ki67+ proliferating cells remained low (Fig 6C). Heatmap shows cytokine protein levels (normalized to vehicle treatment) after 60 d of single agent PD or A8$^{Δ71-73}$. A8$^{Δ71-73}$ + PD0325901 combination treatment or durability treatment shows profound changes in the durability group (Fig 6D). Therapy with MEKi or either C5aRA inhibitor for 60 d resulted in unbalanced changes to the overall cytokine environment, with decreases in the classical pro-inflammatory cytokines IL-23 and TNFa but increases in competing IL-10 (M2, alternatively activated cytokine) and IFNγ (M1, classically activated cytokine). Combination therapy slightly rebalanced the immune microenvironment; the pro-tumorigenic cytokine IL-4

decreased, and pro-inflammatory (M1) cytokine IL-12 increased. Combination therapy also induced a slight increase in RANTES (CCL5). Strikingly, 1 mo after combination therapy in the durability group, the cytokine profile significantly changed. RANTES, which is associated with recruitment of cytotoxic, IFNγ-producing CD4 and CD8 T cells that interact with anti-tumor macrophages, was profoundly up-regulated, and pro-inflammatory (M1) cytokines IL-12 and IFNγ increased. Also, analysis of tissue sections showed that the cellular architecture of these tumors was significantly altered. H&E staining showed cells without signs of atypia or mitosis but with a disorganized architecture. Notable were increased numbers of small round immune cells, present both diffusely and in clusters in all tumor sections from mice in the durability treatment group (Fig 6E).

To characterize the immune cells infiltrating PNF, sections were immostained to detect the myeloid immunomodulation marker CD169 (Siglec-1), which can affect DC and T-cell function. CD169+ cells increased in durability-group tumors versus 60-d-treated tumors (CD169+: 84.23/high powered field (hpf) versus vehicle 25.91/hpf, P = 0.0007) (Fig 6F). Thus, combination of MEK inhibitor with complement modulation caused durable increases in CD169$^+$ macrophages, without re-invigorating tumor Schwann cell proliferation. Macrophages (as defined by robust expression of Iba1), the primary cell type dying in mice treated with the combination therapy, decreased after 2 mo of treatment with PD0325901 (45.35 Iba1+/hpf) or the combination of A8$^{Δ71-73}$ with PD0325901 (36.76 Iba1+/hpf). However, in the durability group, they returned to near-vehicle control counts (79.26 Iba1+/hpf versus vehicle 88.93 Iba1/hpf, P = 0.91, ns) (Fig S4B). Quantification from flow cytometry of MHCII+ myeloid cells (Live, TCRb−, CD3−, CD11b+, MHCII+) and dendritic cells (Live, TCRb−, CD3−, CD11c+, MHCII+) (*P = <0.05, one-way ANOVA) verifies an increase in MHCII high myeloid cells and an increase in CD11c+ DC in the durability setting (Fig 6G). CD3$^+$ T cells are variable and rare in plexiform neurofibroma. CD4 and CD8 T cells slightly decreased as assessed by flow cytometry (Fig S4C and D). Although CD3$^+$ T-cell numbers were marginally increased in the durability group as assessed by counting T cells in tissue sections (CD3+: 11.9/hpf; vehicle 4.8/hpf, P = 0.04), this is likely artifacual because of small round cell clumping (Fig S4E).

To determine if the immunological changes observed in the combination therapy group result in durable effects on tumors at a later time, mice were placed on treatment for 1 mo, at 4–5 mo old, and then maintained off treatment for an additional 4 mo. All tumors grew, whereas off-therapy indicating that growth suppression is not durable (Fig 7A). Remarkably, in contrast, notable changes in the immune microenvironment persisted 4 mo after treatment with C5A antagonist peptide. We observed a significant reduction in total immune cells (Fig 7B) in mice which received either A8$^{Δ71-73}$ alone. In mice which received either A8$^{Δ71-73}$ alone or in combination with MEK inhibitor, there were fewer macrophages as compared with vehicle controls (Fig 7C). The expression of C5aR1 (CD88) on macrophages was unchanged (Fig 7C). We then analyzed macrophage and DC functional markers; we identified a trend toward a reduction in CD169 expression on macrophages and of CD74$^+$ mature dendritic cells. Also, we observed a significant reduction in CD8$^+$ effector T cells (CD8a$^+$CD44$^+$CD62L$^-$) in these tumors (Figs 7C and S5). Therefore, tumor growth suppression is not

**Figure 6. Mice treated with combination therapy for 1 mo and then observed for 1 mo show rebound tumor volume but not increased cell proliferation.**
**(A)** Treatment regimen scheme; in some mice, treatment was discontinued after 1 mo (durability group). Solid line denotes months 7–8, when mice were given combination treatment. Dashed line denotes month 8–9, when mice were observed without treatment before final MRI and tumor excision at month 9. **(B)** Tumor change volume over time for mice treated for 2 mo (PD0325901 and A8$^{\Delta71-73}$ + PD0325901) versus durability group where combination treatment was stopped after 1 mo. **(C)** Quantification of Ki67 immunofluorescence of durability-group tumors compared to tumors with continued treatment (from Fig 4). **(D)** Heatmap shows cytokine protein levels (normalized to vehicle treatment) after 60 d of single agent PD or A8$^{\Delta71-73}$, A8$^{\Delta71-73}$ + PD0325901 combination treatment or durability treatment shows profound changes in the durability group. **(E)** H&E staining of sections from durability-group PNF, demonstrating chaotic architecture with clumps of immune-cell infiltration. Scale bars = 50 $\mu$m. Inset, H&E of vehicle treated tumor. **(F)** Quantification and immunohistochemistry show immunoregulatory macrophages (CD169+) within treatment groups. *$P$ = <0.05; **$P$ = <0.01; ***$P$ = <0.001; ****$P$ = <0.0001 by Tukey's multiple comparisons test. **(G)** Quantification from flow cytometry of MHCII+ myeloid cells (live, TCRb−, CD3−, CD11b+, MHCII+) and dendritic cells (live, TCRb−, CD3−, CD11c+, MHCII+) (*$P$ = <0.05, one-way ANOVA) verifies an increase in MHCII high myeloid cells and an increase in CD11c+ DC in the durability setting.

durable, but single-agent effects of complement inhibition persist for at least several months.

## Discussion

In this study, we sought to improve on MEKi as a treatment for PNF because single-agent MEKi does not kill PNF cells and must be continuously administered for effects to persist. Here, we identified C5a as an anaphalatoxin present at high levels despite MEKi and tested a combination regimen in a preclinical model of PNF. We found that combining a MEK inhibitor with either of two C5aR antagonists caused sustained tumor cell and immune-cell death, and that tumor size decreased on therapy. Based on our results using animals with loss of C5aR1, we anticipate that this combination will have significant local tumor effects with minimal systemic toxicity. Whereas tumors regrew after stopping combination therapy, we demonstrated a sustained effect on the tumor

microenvironment, including rebalancing tumor cytokines and altered localization of CD8$^+$ T cells, emphasizing roles for C5a during early, benign tumorigenesis.

We chose to antagonize C5a receptors, not C5a or total C5, to minimize potential treatment-related toxicity. Each of the three known complement activation pathways converge at the level of C3; proteolytic cleavage of C3 into C3b and C3a forms a C5-cleaving complex that cuts C5 into C5b and C5a. C5b peptide is integral in formation and activation of the membrane attack complex (MAC) (Gunn et al, 2012). Absence of MAC causes immunosuppression, whereas C5aR-specific antagonism will leave the C5b/MAC pathway intact, avoiding significant risk of infection when on therapy.

Whether remaining circulating C5a remains active in the presence of receptor antagonist(s) is unknown. Currently, no direct PD measurement of C5a receptor activation is available for use in vivo. Whereas we developed a novel sandwich ELISA to measure plasma C5a concentrations in mice in hopes that C5a levels might be a biomarker of efficacy, C5a levels trended toward a reduction in treatment, but the effect was not significant. Importantly, 1 mo after

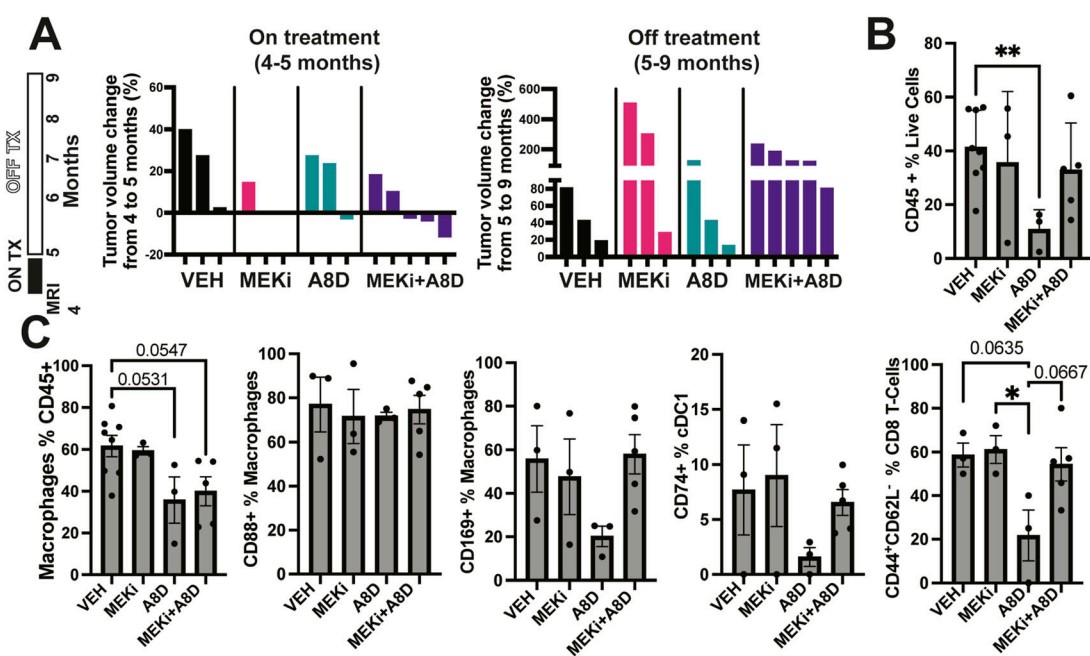

**Figure 7. Mice treated with single-agent therapy for 1 mo and then observed for 4 mo show persistent changes in the immune environment.**
**(A)** Treatment regimen scheme and waterfall plots showing percent change in tumor volume in mice on treatment for month (imaging at 5 mo; left), these same mice were re-imaged 4 mo later, showing tumor growth in all groups (right). **(B)** Quantification of flow cytometetric data, showing reduction in total immune cells (CD45$^+$) in tumors from mice who previously had received A8$^{\Delta71-73}$. **(C)** Quantification of flow cytometetric data showing changes in immune-cell populations and phenotypes after 4 mo off treatment, selective for mice treated with A8$^{\Delta71-73}$; reductions in macrophages were also present in mice treated with the combination.

drug is stopped in our durability treatment, C5a levels were reduced on a cytokine array, suggesting that activation of the C5a pathway is reduced as an anti-tumor environment cytokine/chemokine develops (Fig 6D).

We compared the results of C5aR1-specific PMX205 to those of the dual C5aR1/2 antagonist, A8$^{\Delta71-73}$. The pharmokinetic (PK) properties of PMX have been previously reported (Kumar et al, 2018), but A8$^{\Delta71-73}$ PK properties were unknown. We found that A8$^{\Delta71-73}$ has a very short half-life in plasma (<1 h); levels did not vary between subcutaneous (SQ) versus intraperitoneal (IP) injections (not shown). In contrast, PMX205 showed substantially longer half-life after SQ injection, justifying our use of SQ injection for this agent. Recent evidence suggests specific functions of C5aR2 in response to sterile and bacterial inflammatory danger signals (Breivik et al, 2011; Pandey et al, 2017; Li et al, 2020; Zhang et al, 2020). Given that sterile inflammation is, at least in part, inflammation in many tumors, we postulated that blockade of both C5a receptors might be necessary for maximum efficacy. Despite a much shorter half-life of A8$^{\Delta71-73}$ and the same dosing timeline, PMX205 was not superior to A8$^{\Delta71-73}$. Rather, targeting both C5aR1 and C5aR2 showed modest single-agent effects, whereas targeting C5aR1 alone did not. Targeting both receptors also increased the percent of tumors showing tumor shrinkage in combination with MEK inhibitor. Nonetheless, the inhibitions of C5aR1 versus C5aR1/2 were not statistically different in combination with MEKi. Whereas the two peptides showed similar in vivo effects, in each experiment, A8$^{\Delta71-73}$ trended toward more efficacy, consistent with the idea that targeting both receptors might provide some added benefit. Also, altering dose or schedule of the shorter-lived A8$^{\Delta71-73}$ might improve efficacy.

We detected death of both tumor cells and immune cells after C5aRA treatment, with more cell death in C5aRA plus MEKi combination-treated tumors. Cell death is not detected after MEKi treatment alone. We have not defined the underlying mechanism(s) that cause cell death, but the literature has recently identified C5aR2 as an important activator in inflammasome signaling and tumorigenesis (Karki & Kanneganti, 2019; Yu et al, 2019; Zhang et al, 2020). Moreover, the changes seen in our cytokine array demonstrate increased cytokine/chemokines which stimulate antigen-presenting cells, induce CD8 T-cell migration, and promote cell death. We also detected some dying tumor SCs inside macrophages, in vivo and in vitro, indicating interaction between SCs and macrophages (Figs 3E and 5C). Changes in the tumor microenvironment also directly increase the phagocytic activity of macrophages selectively toward SCs, and blockade of MEK or blockade of C5AR1 increased phagocytosis of Schwann cells in vitro (Fig 5A and B). A straightforward explanation of these results is that C5aR antagonism blocks phagocytosis, enabling visualization of ongoing PNF Schwann cell death. Previous studies have demonstrated the abundance and importance of immune cells in PNF development. However, targeting immune signaling with single-agent therapies has, to date, not proven durable or as successful as MEK inhibition (Prada et al, 2013; Liao et al, 2018; Fletcher et al, 2019). Flow cytometry confirmed C5aR which was largely on PNF macrophages and on other myeloid cells present at lower abundance in PNF, including dendritic cells. This finding is consistent with studies in more aggressive tumors, in which C5aR1 was present in macrophages and dendritic cells (Gerard & Gerard, 1991; Medler et al, 2018). Thus, C5aR on macrophages and DC provided a target in the

PNF tumor microenvironment. In spite of C5aR1 expression, deletion of C5aR1 has no major impact on any major immune subset in these nerve sheath tumors. Proportions of macrophages (resident and infiltrating), MDSCs, dendrtic cells (cDC1 and cDC2), and T cells (CD4 and CD8) were maintained in both genetic models. Interestingly, in the off treatment group, we identified significant changes in the tumor microenvironment, with a decrease in tumor macrophages, reduction in inflammatory dendritic cells, and a significant decrease in CD8 T-effector cells treated with single-agent C5AR inhibition. However, this reduction did not correlate with reduced tumor burden. We are likely insufficiently modulating the immune system using C5aR1 antagonism, based on the discordance between therapeutic response and macrophage burden.

Further characterization of specific myeloid subsets will define their roles in PNF and specific of C5aR antagonists on each cell population. Bulk analysis of PNF macrophages revealed a mixed M1 (anti-tumor)/M2 (pro-tumorigenic) RNA signature, and single cell analysis revealed a subpopulation marked by CC1qa, C1qb, and C1qc, previously identified in tumor-promoting macrophages correlating with T-cell exhaustion (Choi et al, 2017; Kershner et al, 2022; Magrini et al, 2022). Notably, transient treatment of tumors with either C5aRA peptide increased macrophage expression of CD169 (Siglec-1) (Fig 6F). However, 4 mo off drug, CD169 expressuin on macrophages was no longer up-regulated (Fig 7C). CD169 is an adhesion molecule that plays context-dependent roles; it is a macrophage marker distinct from protypical M1/M2 markers, with tumor-associated macrophage immunomodulatory properties (Saito et al, 2015; Fraschilla & Pillai, 2017; Kawaguchi et al, 2022). In mice, CD169 expression induces CD8[+] (cytotoxic) T-cell responses within tumor microenvironments, and CD169 knockout decreases numbers of activated CD8 cells (van Dinther et al, 2019; Grabowskia et al, 2021). Its expression correlates with increased IFN-1 production and thus T-cell activation and with activation or inhibition of cDC1 dendritic cells (Grabowska et al, 2021). In our study, an increase in CD169+ cells correlated with altered localization of CD8[+] T cells and development of an anti-tumorigenic milieu, with persistent reduction in IL-10 and IL-4 and increases in IL-12 and IFNγ in PNF 1 mo after cessation of combination treatment (Fig 6D). By 4 mo off treatment, CD8 effector T cells were reduced (Fig 7C). Therefore, the functions of PNF T cells and their interactions with myeloid cells merit further study.

Sustained environmental response to combination therapy, with tumor growth, is suggestive of delayed responses to direct T-cell modulating agents. For example, biopsies from melanomas from individuals treated with ipilimumab show acute inflammatory reactions similar to those shown here (Wolchok et al, 2009). In human, consecutive re-imaging can distinguish persistent increases in tumor size (progressive disease) from tumors that enlarge transiently and then shrink (pseudo-progression). Pseudo-progression can be caused by continued tumor growth (until an anti-tumor immune response occurs) and/or by transient immune-cell infiltrates that increase tumor cellularity and size. In PNF, tumors decreased in size on therapy, but volumes increased after therapy stopped. However, we did not observe significant tumor cell proliferation, so we favor the interpretation that regrowth occurs because of immune-cell infiltration after stopping therapy (Fig 6F and G). Even 4 mo after stopping treatment, tumors persisted (Fig 7),

yet immune-cell changes persisted, especially after single-agent C5AR antagonist treatment. Thus, there were notable and prolonged changes in tumor microenvironment in the durability study. Timing of drug administration, by blocking MEK to increase C5aR1 expression on immune-cell surfaces and then inhibiting C5aR might improve overall tumor response. Overall, although a promising target initially, our data indicate that as a single agent or combination with MEK inhibition, C5aR antagonism is insufficient to drive significant therapeutic benefit in PNF; thus, alternative combinations or alternative schedules require investigation.

Several considerations suggest that T-cell-targeted therapies may be difficult to use in PNF. For example, CTLA-4 and PD1/L1 immune checkpoint inhibitor (ICI) therapies frequent in clinical use target T cells, but T cells are present in low abundance in PNF (Kershner et al, 2022). Also, immune-related toxicities after ICI therapy may be unacceptable given the generally benign course of PNF. By targeting the tumor-specific macrophages using C5aRA with MEKi, it may be possible to minimize the autoimmune complications of ICIs; C5a up-regulation in PNF is local, so its targeting of macrophages may not induce systemic autoimmune toxicity.

In sum, we identified a tolerable and effective combination therapy in a validated mouse model of neurofibroma. When not statistically better than MEKi alone, the combination of C5aR antagonists with MEK inhibition shows a trend toward increased tumor shrinkage, causes death of tumor cells, and causes long-term effects on the tumor microenvironment including modulation of macrophage phenotype and dendritic cell accumulation. MEK inhibition increases C5aR1 expression on the macrophage cell surface, suggesting that MEK inhibition, followed by C5aR1 inhibition, might provide added benefit and may best be carried out in humans, as PK properties may differ across species. Also, we showed potential benefit to blocking both C5aR1 and C5aR2 receptors in PNF. The anti-human C5/C5a antibody MEDI7814 neutralizes the C5aR1 and C5aR2 receptors (Colley et al, 2018); a C5aR antagonist, Avacopan (CCX168), was approved in 2022 for use in ANCA vasculitis (Jayne et al, 2021). These agents may be of interest to test in human PNF or in >10% of sporadic tumors that show *NF1* mutations.

## Materials and Methods

### Mouse husbandry

All experimental procedures were approved by the Institutional Animal Care and Use Committee (IACUC) at Cincinnati Children's Hospital Medical Center. Mice were housed in temperature- and humidity-controlled facilities with free access to food and water on a 12-h light–dark cycle. Mouse lines used are *Nf1*[fl/fl] (Zhu et al, 2001) and *DhhCre* (Jaegle et al, 2003). The *DhhCre* allele was maintained on the male; *DhhCre;Nf1*[fl/+] mice were crossed to *Nf1*[fl/fl] to obtain tumor-bearing mice. We bred a β-actin flox/stop/flox-EGFP mouse reporter line in which the CMV-b actin promoter and loxP-flanked CAT gene are upstream of the enhanced green fluorescent protein cassette (Nakamura et al, 2006). This reporter was crossed into the *Nf1*[fl/fl] background to obtain the EGFP–;*Nf1*[fl/fl] mice. All mice were

maintained on a C57BL/6 background. *Nf1* genotyping was performed using the oligonucleotides (5′ to 3′) CTT CAG ACT GAT TGT TGT ACC TGA and ACC TCT CTA GCC TCA GGA ATG A to detect the WT allele, and TGA TTC CCA CTT TGT GGT TCT AAG to detect the targeted allele. *DhhCre* genotyping was performed using the forward primer ACCCTGTTACGTATA GCCGA and reverse primer CTCCGGTATTGA AACTCCAG. GFP genotyping was performed using the forward primer ACGTAAACGGCCACAAGTTCA and reverse primer GCTGTTGTAGTTG-TACTCCAGGT. C5aR1−/− mice were a gift from Dr. M. Pandey (Höpken et al, 1996) on a C57BL/6-BALB/c-129 mixed genetic bacground. C5ar1 genotyping was performed using the following primers (5′–3′): WT forward: GGTCTCTCCCCAGCATCATA, mutant forward: GCCA-GAGGCCACTTGTGTAG, and common: GGCAACGTAGCC AAGAAAAA. WT and common primers detect WT band, mutant, and common primers detect mutant band.

## Drug dosing

PD0325901 was synthesized by Selleck, and A8$^{\Delta71-73}$ was synthesized by GenScript. PMX205 was provided by Professor Trent Woodruff. Mice were administered vehicle (0.5%) (wt/vol) methylcellulose solution with 0.2% (vol/vol) polysorbate 80 (Tween-80) for oral gavage, 5% glucose water subcutaneous, or PBS intraperitoneal for vehicle control. Mice were administered medication once daily for 5 d on, 2 d off. PD0325901 was dissolved in methylcellulose solution with 0.2% (vol/vol) polysorbate 80 (Tween-80), and 1.5 mg/kg/d was administered by oral gavage. PMX205 was dissolved in 5% glucose [w/w] water, and 1 mg/kg was injected subcutaneously. A8$^{\Delta71-73}$ was dissolved in PBS and 5 mg/kg admistered by intraperitoneal injection. Mice were treated for 5–60 d, as noted in the text.

## MRI imaging

MRI was performed on anesthetized *Nf1$^{fl/fl}$;DhhCre* mice on a 7T Bruker Biospec, and tumor volume was quantified as described (Wu et al, 2012; Jessen et al, 2013). Mice were scanned at 5, 7, and 9 mo of age (m), with some instances, as noted in the text, an additional MRI scan at 8 mo.

## Plasma and tumor collection, processing, and storage

2 h after the last treatment dose (unless specified), mice were anesthetized with isofluorane, and blood was collected by cardiac puncture. Blood samples were transferred to anti-clotting EDTA tubes on ice and centrifuged within 30 min of collection at 4°C for 15 min at 14,000g. Plasma was transferred to 1.5 ml Eppendorf tubes on ice and stored at −80°C until analysis. After cardiac puncture, we euthanized mice and dissected paraspinal PNF within 10 min and flash frozen in liquid nitrogen, then stored in −80°C until analysis, or fixed in 4% PFA in PBS for 1 h at RT, then placed into 20% sucrose overnight at 4°C. The next day, tissue was embedded into optimal cutting temperature (O.C.T.) embedding medium (4585; Thermo Fisher Scientific), and blocks were stored at −80°C until sectioning.

## Tissue sections, histology, and immunofluorescence

Frozen cryosections (12 *μ*m) were dried and stored at −80°C. Slides were fixed with 4% PFA in PBS for 10 min, washed with PBS or tris-buffered saline with 0.1% Tween (TBST) for 5 min, thrice. For immunofluorescence, slides were blocked in 10% normal goat serum in TBST for 30–60 min and rinsed with TBST thrice, then incubated in anti-Cleaved Caspase 3 (rabbit, 96661S, 1:400; Cell Signaling), anti-CD11b (rat, 553308, 1:250; BD Pharminogen), anti-2',3'-cyclic nucleotide 3'-phosphodiesterase (CNPase, chicken, AB9342, 1:250; Millipore), or anti-Ki67 (rabbit, 12202S, 1:200; Cell Signaling) diluted in blocking medium overnight at 4°C. Sections were rinsed with TBST or PBS x3 and incubated with an appropriate secondary antibody at 1:500 (Alexa Fluor). Slides were finally rinsed, incubated with 1:10,000 DAPI rinsing, and coverslipped in Fluoromount G (17984-25; Electron Microscopy Sciences). Fluorescent images were captured using a Zeiss Axiovert 200M fluorescent microscope with 405, FITC, Cy5, and Texas Red filters. Quantification of cells was carried out in 5–10 HPFs (400x)/tumor. Immunohistochemistry staining followed the above protocol with the addition of 10% hydrogen peroxide quenching to block endogenous peroxidase before blocking with normal goat serum. Slides were stained with H&E (Richard-Allan Scientific Series Hemotoxylin 1 7221 and Richard-Allan Scientific Eosin-Y 7111; Thermo Fisher Scientific), anti-S100 (Z0311, 1:10,000; Dako), Ki67 (12202, 1:250; Cell Signify), Iba1 (019–1974, 1:3,000; WAKO), CD3 (100202, 1:500; BioLegend), CD8 (10301, 1:500; Invitrogen), CD169 (142402, 1:250; BioLegend). Human parafin sections underwent antigen retrieval with citrate buffer (pH 6.0) prior to staining as per the above protocol for C5aR1 (LS-C358932, 1:400; LifeSpan BioSciences), anti-S100 (Z0311, 1:10,000; Dako), and H&E (Richard-Allan Scientific Series Hemotoxylin 1 7221 and Richard-Allan Scientific Eosin-Y 7111; Thermo Fisher Scientific). Images were captured with a SPOT Insight 4 Mp CCD camera (Spot Imaging). Quantification of cells was counted in an average of 5–10 HPFs (40x)/tumor.

## TUNEL staining

The TUNEL assay was performed, according to Roche in situ cell death detection kit and TMR red instructions (Roche Diagnostics Gmbh, Cat #: 12156792910; Sigma-Aldrich) on cryosections. DAPI was used to stain nuclei. TUNEL+ cells and DAPI+ nuclei were counted in at least three cross-sections per sample. Data are presented as average percentage of TUNEL+ cells per sample.

## Cell preparation and flow cytometry

DRG or paraspinal tumors were dissected from intracardially ice-cold PBS-perfused animals, cut finely, and incubated in disso-ciation media containing RPMI 1640 (11875-119; Invitrogen), 10 mg/ml Collagenase A, from Clostridium histolyticum (cat# 10103586001; Sigma-Aldrich), 10 mg/ml Collagenase, Type 4 (cat # LS004188; Worthington Biochemical), 100 *μ*g/ml Trypsin Inhibitor, from soybean (cat # 10109886001; Sigma-Aldrich), 250 U DNase I (cat#10104159001; Sigma-Aldrich) 5 *μ*M CaCl$_2$ at 37°C for 30 min. Samples were then mechanically dissociated by gentle pipetting, washed two times with 25 ml of RPMI containing 10% FBS, and filtered through first a 100 and then a 70 *μ*M cell trainer before being resuspended in standard flow buffer. All subsequent steps

were performed on ice. After labeling with live-dead exclusion marker (Live/Dead Blue, Fisher) per manufacturer's protocol, tumors were washed and resuspended with 1 ml of Flow Buffer (BioLegend). A staining cocktail containing Brilliant Buffer (1:200; BD Bioscience), True-Stain monocyte block (1:200; BioLegend), Cd16/Cd32 rat anti-mouse 2.4G2 block (1 $\mu$M; BD Bioscience), and primary antibodies (Fig S2A) was incubated with cells for 30 min. After incubation, cells were washed twice and fixed with 2% fresh PFA (30 min at RT). For panels with nuclear markers, samples were permebalized with TrueNuclear Permeabilization Buffer (BioLegend) and labled as described above with antibodies detecting intracellular targets. Samples were run on either a Cytek Aurora spectral analyzer (Cytek) or MageStreamX Mark II (Sigma-Aldrich) within 72 h. Single-color controls and unstained controls were used to unmix samples. Representative gating of antibody panel 1 is shown in Fig S2B.

### In vitro phagocytosis assay

BMDM were generated from male and female $Nf1^{fl/fl};DhhCre^+$ tumor-bearing mice (6–9 mo of age). Whole blood, femur, and tibia were extracted after euthanasia. Serum was extracted from whole blood via centrifugation. Bone marrow was flushed using Opti-MEM supplemented with GlutaMAX (Thermo Fisher Scientific), mechanically dissociated, filtered through a 40-micron cell strainer, and plated onto 12-mm-diameter acid-washed glass coverslips placed into wells of 24-well tissue culture plates. Non-adherent cells were washed away with warmed medium after 2 h, and cells were maintained in Opti-MEM supplemented with GlutaMAX, 1% autologous serum, and 1% penicillin/streptomycin for 6 d. BMM were switched to serum-free medium 48 h before functional assays. Mouse Schwann cells were isolated from the DRG of $Nf1$-null mutant mice at embryonic day 12.5 as previously described (Kim et al, 1997). Neurons and fibroblasts were removed from cultures, cells maintained for 7 d, then trypsinized and replated in a six well dish and expanded until confluent in Schwann cell growth medium containing 2 $\mu$M forskolin (Cayman Chemical Company), 10 ng/ml recombinant human NRG1-beta 1/HRG1-beta 1 EGF Domain peptide (Cat# 396-HB-050; R&D Systems), and 10% FBS. Schwann cells were labeled with 5 $\mu$m of CFSE (BioLegend) at 37°C for 20 min, thoroughly washed with PBS, and incubated in warmed Opti-MEM for 30 min. BMM were treated with PD0325901 (1 $\mu$M), A8$\Delta^{71-73}$ (1 $\mu$M), or vehicle control for 8 h. Schwann cells were trypsinized, washed, and 8 × 10$^4$ labeled Schwann were added to pre-treated BMM. After 3 h of incubation, cells were quickly washed and fixed with 2% PFA. Coverslips were labeled with anti-SIRP$\alpha$/CD172a (144001, 1:1,000; BioLegend) and counterstained with DAPI before being imaged on a Nikon C1 confocal (Nikon Instruments) under identical laser power, gain, and pixel dell. Macrophages were identified using CD172a reactivity, and mean intensity of CFSE within identified macrophages was measured using Cell Profiler and ImageJ software. Nucleated Schwann cells were excluded from analysis.

### In vitro phagocytosis assay with isolated tumor cells

PNF from mice used to prepare BMDM were collected and dissocated as described for flow cytometry, and 5 × 10$^6$ cells from each tumor labled with either 0.1 $\mu$M of CFSE Cell Division Tracker Kit (BioLegend) or CellTracker Violet BMQC Dye (5 $\mu$M; Thermo Fisher Scientific) for 15 min on ice. Cells were then washed twice, combined, and cultured in ultra-low binding round bottom plates for 8 h at 37°C and 5% $CO_2$ in OPTI-MEM plus 1% PenStrep without added serum. After incubation, cells were removed and flow cytometry was used.

### Cytokine array

Cytokine arrays were performed according to mouse cytokine array kit (mouse cytokine array panel A, cat# ARY006; R&D Systems). 300 $\mu$g total protein was loaded into each membrane. The relative intensities of the black dots were measured using film or an Azure Biosystems c600 imager and quantified using ImageJ software. Relative protein expression is shown in each experiment for proteins expressed in at least one condition in that experiment.

### Western blot

Western blots were performed using antibodies from Cell Signaling Technology recognizing PARP (1:1,000, Cat # 9542) and $\beta$-actin (1:10,000, Cat# 5125S) on 4–20% gels transferred onto nitrocellulose membranes.

### C5a ELISA

We coated white 96-well plates with purified rat anti-mouse C5a mAb (#558027; BD) at 2 $\mu$g/ml in 1X Tris Saline (pH 7.2) and incubated plates overnight at 4°C. We washed plates 6 times with 1X Tris saline and 0.05% Tween 20 (WB). We added serially diluted standards (mouse recombinant C5a, cat# HC1101; Hycult Biotech) and samples (plasma or tumor lysates) to corresponding wells and incubated plates for 2–3 h at RT. 50 $\mu$l/well biotin Rat anti-mouse C5a mAb (#558028; BD) were added to each well at a 1:5,000 dilution in dilution buffer (10% SuperBlock: cat#37545; Fisher/Thermo/Pierce) and incubated plates at RT for 25 min. We next added 50 $\mu$l Streptavidin-HRP (21126, 1:20,000; Fisher/Thermo/Pierce) to each well and incubated plates at RT for 25 min. Plates were washed six times in WB between each step above. We then added 150 $\mu$l of freshly prepared SuperSignal Elisa substrate (1 ml $H_2O_2$ + 1 ml Enhancer + 15 ml 1X Tris Saline) to each well and mixed well. We read the values immediately using a 96-cell Luminometer reader.

### PKs

Plasma was stored at −80°C. On the day of analysis, after thawed, 40 $\mu$l of each plasma sample, reference standard, or quality control was precipitated by the addition of 40 $\mu$l of isopropanol containing 0.1% formic acid. After vortex and centrifugation, 50 $\mu$l of supernatant was transferred into a new plate and diluted with 100 $\mu$l 0.1% formic acid in water. 5 $\mu$lL of each sample was injected for analysis by LC/MS. Samples were analyzed by a high-resolution orbitrap mass spectrometer Q-Exactive (2012; Thermo Fisher Scientific) coupled with Accela 1250 HPLC system using SIM scan mode. An expected isotope accurate m/z list (1015.1459, 1015.2706, 1015.3949, and 1015.5187) for C5aRA was included. The mass spectra were

collected at a resolution of 70,000 with the AGC target of $1.00 \times 10^5$, maximum IT of 100, and an isolation window of 4 m/z. To separate C5aRA from the interference signals, the samples were injected onto an Agilent peptide column (AdvancedBio Peptide C18; 2.1 × 150 mm, 2.7 $\mu$m) and chromatographed using a 15-min gradient with 0.1% formic acid in water and 0.1% formic acid in isopropanol/acetonitrile mixture. Thermo Xcalibur (V3.1) software was used for data acquisition and processing. Samples were quantified against a standard curve spiked in the blank plasma at 25–10,000 ng/ml, along with the quality controls spiked at QC-low (50 ng/ml), QC-med (250 ng/ml), and QC-high (2,500 ng/ml) levels at Paraza Pharma.

### Human gene expression data

We re-analyzed published gene expression microarray data (GEO accession: GSE14038, Affymetrix GeneChip HU133 Plus 2.0), including 10 normal human Schwann cell (NHSC), 11 plexiform neurofibroma Schwann cell, 3 nerve, and 13 PNF samples. Probe sets were summarized by the robust multi-chip average (RMA) method in Bioconductor Affy package. The heatmap represents z-scores across samples.

### Statistical analysis

Tumor shrinkage analsysis and random coefficient model analysis were completed as described in Wu et al (2012). All other statistical analyses were conducted in GraphPad Prism. Significance was set at $P \leq 0.05$. Statistical tests and any posthoc tests used for each experiment are given in figure legends. Error bars represent the SEM, unless otherwise noted.

## Supplementary Information

## Acknowledgements

This work was supported by NIH R61/33 NS112407 to J Wu and N Ratner.

## Author Contributions

MR Perrino: data curation and writing—original draft.
N Ahmari: data curation, formal analysis, and writing—original draft.
A Hall: data curation.
M Jackson: data curation.
Y Na: data curation.
J Pundavela: data curation.
S Szabo: data curation and investigation.
TM Woodruff: resources and methodology.
E Dombi: formal analysis.
M-O Kim: formal analysis.
J Köhl: conceptualization and investigation.
J Wu: conceptualization, data curation, formal analysis, supervision, funding acquisition, project administration, and writing—review and editing.
N Ratner: conceptualization, supervision, funding acquisition, project administration, and writing—review and editing.

### Conflict of Interest Statement

The authors declare that they have no conflict of interests.

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
