## [Reviewer comments · Life Science Alliance]

Life Science Alliance

C5aR plus MEK inhibition durably targets the tumor milieu and reveals tumor cell phagocytosis

Melissa Perrino, Niousha Ahmari, Ashley Hall, Mark Jackson, Youjin Na, Jay Pundavela, Sara Szabo, Trent Woodruff, Eva Dombi, Mi-Ok Kim, Jörg Köhl, Jianqiang Wu, and Nancy Ratner

DOI: <https://doi.org/10.26508/lsa.202302229>

Corresponding author(s): Nancy Ratner, Cincinnati Children's Hospital Medical Center

Review Timeline:

Submission Date:	2023-06-21
Editorial Decision:	2023-08-21
Revision Received:	2024-02-05
Editorial Decision:	2024-02-09
Revision Received:	2024-02-13
Accepted:	2024-02-14

Transaction Report:

August 21, 2023

Re: Life Science Alliance manuscript #LSA-2023-02229-T

Dr. Nancy Ratner
Cincinnati Children's Hospital Medical Center
Experimental Hematology and Cancer Biology
3333 Burnet Avenue
Cincinnati, Ohio 45229-0713

Dear Dr. Ratner,

Thank you for submitting your manuscript entitled "C5aR antagonism together with MEK inhibition durably targets the tumor microenvironment and reveals ongoing tumor cell apoptosis" to Life Science Alliance. The manuscript was assessed by expert reviewers, whose comments are appended to this letter. We invite you to submit a revised manuscript addressing the Reviewer comments.

Thank you for this interesting contribution to Life Science Alliance. We are looking forward to receiving your revised manuscript.

Sincerely,

B. MANUSCRIPT ORGANIZATION AND FORMATTING:

Reviewer #1 (Comments to the Authors (Required)):

In this study, Perrino et al investigated the effects of C5aR deficiency or inhibition in combination with MEK inhibition in mouse model of plexiform neurofibromas. They observed that the combination of MEK and C5aR inhibition did not improve the tumor control induced by MEK inhibition alone. However, they observed that C5aR inhibition induced macrophage and SC apoptosis, and other modifications of the tumor microenvironment, including cytokines. Unfortunately, the relevance of these findings is not explored in the study.

The study is of potential interest since this tumor is highly infiltrated by C5aR1+ macrophages which may be reactivated in an antitumor phenotype upon C5aR inhibition.

However, most results are not explained mechanistically, thus strongly reducing the interest and soundness of the study. Conclusions are not supported by solid experimental evidence. For instance, how is C5aR inhibition inducing macrophage and SC apoptosis? And macrophage-mediated phagocytosis of SC? Are the alterations of the tumor microenvironment observed upon combined treatment relevant in terms of tumor growth and anti-tumor immune responses?

Reviewer #2 (Comments to the Authors (Required)):

The authors have identified C5a as a potential target in Plexiform Neurofibromas. Using appropriate genetic and pharmacological inhibition models, the authors have demonstrated that C5aR surface presentation is enhanced in the PNFs and further accentuated in the settings of MEK inhibition. The authors have shown that ablating or inhibiting C5aR leads to enhanced Macrophage and Schwann cell apoptosis and increased Macrophage phagocytosis of Schwann cells. Combined MEK inhibition and C5aR antagonism bring sustained alteration of the tumor microenvironment.

This study offers C5aR as a potential new druggable target in the PNFs, for which no durable cure exists. As such, the study is translationally relevant and warrants dissemination to the scientific community. However, I have a few comments to strengthen the output of the manuscript.

1. While the impact of C5aR antagonism on the TME is evident, it needs to be clarified why there was no treatment benefit of the combination treatment with MEKi. The authors have shown that the combination is not toxic and well tolerated; however, if there is no benefit to the tumor outcome, there would be no point in proposing this treatment. How do the authors address this concern?

2. The authors have shown some signs of the durability of the TME-altering response in the combination settings. However, as they suggested, with a longer-term tumor study, it is possible to determine whether the tumor will shrink or continue to enlarge after the treatment. I recommend the authors to address this.

3. In a few instances throughout the manuscript, the authors have stated facts in a way that, while NOT FALSE, could BE Misleading to the readers. For example, on page 17, the title of the last paragraph is "Longer term treatment of C5aRA+MEKi reduces tumor size". While this statement is factually absolutely correct, the true message that should have been communicated was that the combination is NOT doing any better than the MEKi alone. After all, the authors are not trying to demonstrate the impact of MEKi in this manuscript but rather the benefit of the combination. The authors must modify this and others (wherever applicable) to reflect the true essence of their experimental data.

4. On page 3, line 60: it would be "tumor mass" not "tumor cells"

5. Page 16, line 360: it would be "...significantly increased apoptotic cell..." not "...significantly increased dead cell..."

Re: Life Science Alliance - Editorial Decision LSA-2023-02229-T

We thank the reviewers and the editor for their comments on our manuscript “C5aR-antagonism in combination with MEK inhibition durably targets the tumor micro-environment and reveals ongoing tumor cell phagocytosis”, by Perrino, Ahmari and co-workers. You invited us to submit a revised manuscript addressing the Reviewer comments (8/21/23). We appreciate that this revision took longer than your typical 3-month turnaround, but Reviewer 2 requested an in vivo study that required significant analysis. We addressed the reviewer's comments point by point and highlighted the changes in yellow in the manuscript. We hope that you will find this revised manuscript appropriate for publication in Life Science Alliance.

Reviewer #1

In this study, Perrino et al investigated the effects of C5aR deficiency or inhibition in combination with MEK inhibition in mouse model of plexiform neurofibromas. They observed that the combination of MEK and C5aR inhibition did not improve the tumor control induced by MEK inhibition alone. However, they observed that C5aR inhibition induced macrophage and SC apoptosis, and other modifications of the tumor microenvironment, including cytokines. Unfortunately, the relevance of these findings is not explored in the study. The study is of potential interest since this tumor is highly infiltrated by C5aR1+ macrophages which may be reactivated in an antitumor phenotype upon C5aR inhibition. However, most results are not explained mechanistically, thus strongly reducing the interest and soundness of the study. Conclusions are not supported by solid experimental evidence.

For instance, how is C5aR inhibition inducing macrophage and SC apoptosis? And macrophage-mediated phagocytosis of SC?

Response:

This is indeed an important question and indeed, a question we have tried to address. We show using mutant mice that phagocytosis by tumour macrophages is dependent on C5aR and independent of MEK inhibition.

We do not yet know what causes cell death, but we characterized cells which die in tumors and show that at least dying Schwann cells are phagocytosed by macrophages in a MEK or C5aR inhibitor-dependent fashion (Figure 5).

To make these points more clearly, we thoroughly revised Figure 5 and the accompanying text (Pages 19-20).

Are the alterations of the tumor microenvironment observed upon combined treatment relevant in terms of tumor growth and anti-tumor immune responses?

Response:

We spent considerable effort to answer this important question, and the data is presented in a new Figure 7. The answer is no, the combined therapy does not cause durable effects on tumor growth. Regarding immune cells, despite the abundance of C5aR1+ myeloid cells in PNF, very few changes occur on dendritic cells or macrophages on blocking this receptor. However, in the off-treatment groups, we identified significant changes in the tumor microenvironment that persist for at least 4 months, especially after single agent C5aR antagonism (new Figures 7B & 7C).

Reviewer #2 (Comments to the Authors (Required)):

The authors have identified C5a as a potential target in Plexiform Neurofibromas. Using appropriate genetic and pharmacological inhibition models, the authors have demonstrated that C5aR surface presentation is enhanced in the PNFs and further accentuated in the settings of MEK inhibition. The authors have shown that ablating or inhibiting C5aR leads to enhanced Macrophage and Schwann cell apoptosis and increased Macrophage phagocytosis of Schwann cells. Combined MEK inhibition and C5aR antagonism bring sustained alteration of the tumor microenvironment.

This study offers C5aR as a potential new druggable target in the PNFs, for which no durable cure exists. As such, the study is translationally relevant and warrants dissemination to the scientific community. However, I have a few comments to strengthen the output of the manuscript.

1. While the impact of C5aR antagonism on the TME is evident, it needs to be clarified why there was no treatment benefit of the combination treatment with MEKi. The authors have shown that the combination is not toxic and well tolerated; however, if there is no benefit to the tumor outcome, there would be no point in proposing this treatment. How do the authors address this concern?

Response:

We completely agree with this reviewer. Although a promising target initially, our data indicate that as a single agent or combination with MEK inhibition C5aR antagonism is insufficient to drive therapeutic benefit; combinations or alternatives will be necessary. Given the importance of this idea, we have added following sentence to the discussion “Even four months after stopping treatment, tumors persisted (Figure 7), yet immune cell changes persisted. There was a notable and prolonged change in tumor microenvironment in the durability study. These effects best correlated decreased numbers of CD8 T cells..... “Overall, although a promising target initially, our data indicate that as a single agent or combination with MEK inhibition, C5aR antagonism is insufficient to drive therapeutic benefit in PNF; thus, additional combinations or alternative schedules require investigation.” (Page 28).

2. The authors have shown some signs of the durability of the TME-altering response in the combination settings. However, as they suggested, with a longer-term tumor study, it is possible to determine whether the tumor will shrink or continue to enlarge after the treatment. I recommend the authors to address this.

Response:

We agreed with this reviewer and had initiated a longer durability study before manuscript submission. We show that data as an entirely new Figure 7. We found that tumors do not

shrink after transient drug exposure. However, A8^{Δ71-73}-driven C5AaR1 antagonism is sufficient to modulate the PNF tumor microenvironment for months (Pages 22-23).

3. In a few instances throughout the manuscript, the authors have stated facts in a way that, while NOT FALSE, could BE Misleading to the readers. For example, on page 17, the title of the last paragraph is "Longer term treatment of C5aRA+MEKi reduces tumor size". While this statement is factually absolutely correct, the true message that should have been communicated was that the combination is NOT doing any better than the MEKi alone. After all, the authors are not trying to demonstrate the impact of MEKi in this manuscript but rather the benefit of the combination. The authors must modify this and others (wherever applicable) to reflect the true essence of their experimental data.

Response:

We replaced this section heading to read: "C5aRA + MEKi reduces tumor size, similar to MEKi alone." (Page 17)

4. On page 3, line 60: it would be "tumor mass" not "tumor cells"

Response:

We did not change this, as the tumor mass is contributed by tumor cells, immune cells, stromal cells, and matrix, and because 30% of cells are macrophages, as a percent of all tumor cells.

5. Page 16, line 360: it would be "...significantly increased apoptotic cell..." not "...significantly increased dead cell..."

Response:

Given that recent literature finds that not only apoptotic cell death but also other types of cell death can result in cleavage of DNA by Casp3 (e.g. pyroptosis) we prefer this nomenclature. This now reads: "At this early time point, MEK inhibition alone had little effect on death of cells isolated from murine PNF. In contrast, genetic deletion of C5aR1 significantly increased CC3+ cells (Fig. 3C), and the combination of C5aR1 deletion and MEK inhibition did not alter the amount of cell death" (Page 16).

Additionally, we modified figure 2H by adding more time points and sample numbers. We reanalyzed the data using two-way ANOVA, and modified supplemental figures 4 and 5.

Thanks for your consideration.

Sincerely Yours,

Nancy Ratner, Ph.D.
Division of Experimental Hematology
Children's Hospital Research Foundation
Cancer Blood Disease Institute
3333 Burnet Avenue, MLC 7013
Cincinnati, OH 45229-3039, USA
Jianqiang Wu Phone: (513) 636-0955| Fax: (513) 636-3549
Nancy Ratner Phone: (513) 636-9469| Fax: (513) 636-3549
E-mail: Jianqiang.wu@cchmc.org, Nancy.ratner@cchmc.org

February 9, 2024

RE: Life Science Alliance Manuscript #LSA-2023-02229-TR

Dr. Nancy Ratner
Cincinnati Children's Hospital Medical Center
Department of Pediatrics
3333 Burnet Avenue
Cincinnati, Ohio 45229-0713

Dear Dr. Ratner,

Thank you for submitting your revised manuscript entitled "C5aR plus MEK inhibition durably targets the tumor milieu and reveals tumor cell phagocytosis". We would be happy to publish your paper in Life Science Alliance pending final revisions necessary to meet our formatting guidelines.

- please be sure that the authorship listing and order is correct
- please add ORCID ID for the corresponding author -- you should have received instructions on how to do so
- please add the Twitter handle of your host institute/organization as well as your own or/and one of the authors in our system
- please use the [10 author names et al.] format in your references (i.e., limit the author names to the first 10)
- please add an Author Contributions section to your main manuscript text
- please add a Conflict of Interest statement to your main manuscript text
- please move your main and supplementary figure legends to the main manuscript text after the references section
- there is a callout in the manuscript text for Fig. 3SC, and the figure has only panels A and B; please correct
- figure S5 has only one panel; please remove A from the actual figure and correct callouts in the manuscript text
- please add callouts for Figures 6F; S2B; S3A-B; S4B to your main manuscript text

Figure Checks:

- please add sizes next to the blots in Figure 3B

A. FINAL FILES:

B. MANUSCRIPT ORGANIZATION AND FORMATTING:

Sincerely,

Reviewer #2 (Comments to the Authors (Required)):

The authors have reasonably addressed all the concerns raised by this reviewer.

February 14, 2024

RE: Life Science Alliance Manuscript #LSA-2023-02229-TRR

Dr. Nancy Ratner
Cincinnati Children's Hospital Medical Center
Department of Pediatrics
3333 Burnet Avenue
Cincinnati, Ohio 45229-0713

Dear Dr. Ratner,

Thank you for submitting your Research Article entitled "C5aR plus MEK inhibition durably targets the tumor milieu and reveals tumor cell phagocytosis". It is a pleasure to let you know that your manuscript is now accepted for publication in Life Science Alliance. Congratulations on this interesting work.

DISTRIBUTION OF MATERIALS:

Again, congratulations on a very nice paper. I hope you found the review process to be constructive and are pleased with how the manuscript was handled editorially. We look forward to future exciting submissions from your lab.

Sincerely,
